# Role Assignment Mechanism of Unmanned Swarm Organization Reconstruction Based on the Fourth Directed Motif

**DOI:** 10.3390/s22228799

**Published:** 2022-11-14

**Authors:** Ting Duan, Weiping Wang, Tao Wang, Xiaobo Li

**Affiliations:** College of Systems Engineering, National University of Defense Technology, Changsha 410073, China

**Keywords:** swarm reconstruction, motif evolution, flexible role assignment, flexible mechanism design, unmanned swarm mission planning

## Abstract

With the rapid development of intelligent unmanned technology, unmanned combat swarms are faced with a highly aggressive, highly uncertain, and highly dynamic battlefield environment, and the operation mode of unmanned combat has gradually shifted from single-platform operations to swarm networking collaboration combat development. Aiming at the typical characteristics of the unmanned swarm combat system, this paper proposes a role assignment model for organizational reconfiguration at the swarm layer and builds an unmanned swarm organization reconfiguration role-assignment mechanism model (SORAM) based on the fourth-order directed motif. The method starts from the organizational domain of the swarm system and takes the task as the the dependent variable of the role assignment of the swarm organization, quantifies the importance of the motif from a statistical point of view, and establishes a multi-objective model considering the similarity of the structure. The swarm reconfiguration role optimization method of SR-NSGA-2 provides a reference for the online adaptation of the swarm links. Finally, combined with a simulated combat simulation case, the usability and effectiveness of the method are tested.

## 1. Introduction

Unmanned swarm is a mobile multi-agent system which is composed of a certain amount of unmanned combat equipment and has self-organizing characteristics based on a sympathetic network. The reconfiguration of swarm organization is a process of comprehensively considering the units, system disturbance factors, operational layout, and overall operational planning that affect the execution of tasks, starting from the combat effectiveness and mission execution degree of the swarm system. The mission planning of unmanned swarm cannot be separated from the scientific system-of-systems (SoS) architecture design. The SoS warfare makes the future war more and more complex, and the system is “the integration of a limited number of component systems, which are independent and operational, and connected to each other for a period of time to achieve a higher goal”. The complexity of war is difficult to describe, and architecture provides a good way to solve this problem.

The most famous is the research by Madni of the University of California, which mainly focuses on the flexible design of the system architecture [1,2,3,4,5]. He proposed a framework based on contract design and combined with hidden Markov modeling to effectively solve the indescribability of disturbance factors in the system. The International Conference on the Industrial and Production Engineering-SARC seminar held in Venice discussed the robustness and collaboration of the whole multi-agent system in the context of self-organizing and adaptive systems, which provided a lot of the foundation for the study of system resilience. Wang Weiping’s team has also conducted a lot of research on system architecture. They have studied the combat system [6] from various aspects of the design, simulation, evaluation, and optimization of the system architecture, including the research on the mission planning of the UAV swarm driven by the architecture, and proposed the OSoSA search framework. In addition, it is transformed into a dynamic programming problem [7] under the constraint of the uncertainty of the potential capability of the architecture. Considering the possibility of UAV failure, this paper studies the task scheduling problem of UAV swarm based on adaptive network mapping to the architecture [8]. Swarm organization reconfiguration belongs to the field of unmanned swarm task planning. The existing research in this field mainly focuses on improving the performance of tasks, such as cooperative control, formation control, path planning, and task allocation. The above research generally assumes that the unmanned combat equipment is in a relatively static environment by default. However, in the actual combat environment, disturbance factors exist, including external real-time and variable environmental situation disturbance and internal component fault communication interruption. The disturbance will cause node failure and link interruption of the unmanned swarm network, which will lead to the change of swarm network topology and affect the combat efficiency of the whole system. Therefore, for architecture design researchers, the close cooperation of swarm layer effectively improves the coupling of system layer, and the organizational reconfiguration of swarm layer is a very important research topic.

In addition, cluster planning is now a multi-objective optimization problem. A lot of goals will pick up with the search for the Pareto frontier, making increased computational complexity, a general artificial immune system, a large number of antibodies, through the local information exchange, reaching the target of overall coordination, not under central control [9], and our mission planning problem belongs to a small amount of centralized distributed programming, not suitable for direct application of this method. The particle swarm optimization algorithm will fall into the optimal solution prematurely, while our scheduling process is a variable and continuous process [10]. If there are commander nodes, alternative solutions may be directly selected. The plasticity of the NSGA-2 algorithm is strong, which can ensure that individuals with good characteristics can remain in the population [11,12]. Therefore, after the cluster role reconstruction algorithm, we apply the NSGA-2 algorithm for optimization.

Some studies have begun to pay attention to the method of UAV swarm coping with disturbance, and put forward relevant design indicators. Ordoukhanian uses the flexible UAV swarm system to consider the cost of disturbance factors and establish a cooperative multi-objective function [13]. This paper introduces the destructibility into the multi-agent system and carries out the destructibility design [14,15,16] by two standards of K-connectivity and system energy consumption. Later, the concept of resilience is gradually applied to the research of UAV swarm, which can measure the whole process from damage to recovery and comprehensively evaluate the ability to resist interference in the process of combat [17]. However, most of the existing research considers the survivability and adaptability design of the system from the overall combat system layer but does not consider the adaptability design from the positive perspective of the swarm layer organization reconstruction.

The main contributions of this article can be summarized as follows:

1. By using the characteristics of the motif, this paper introduces the fourth-order motif into the cluster architecture, and proposes and establishes the cluster reconstruction model for the first time.

2. In order to reflect the ability of the motif, the fourth-order motif model was quantified, and three evaluation indexes of the motif structure similarity, importance, and connection time were established.

3. The cluster reconstruction algorithm in the cluster role reconstruction mechanism was established;

4. In order to find the high-quality optimal solution quickly, the authors of this paper thought of combining the improved genetic algorithm after reconstruction to find the solution set quickly. After comparing with some algorithms, it is found to be effective.

The following arrangement of the paper is as follows Figure 1: Section 2 firstly establishes a fourth-order motif model of role reconstruction (M-SAR2) from the perspective of the motif, which is used as the basis of the research on the role allocation mechanism of swarm organization reconstruction. In Section 3, the analysis index of swarm organization structure is proposed, then the index of dynamic reconstruction of the fourth order motif is established. In Section 4, the role assignment mechanism algorithm of swarm reconstruction is given, the optimization problem model is established for the role assignment index, the improvement is based on NSGA-2 algorithm, and the SR-NSGA-2 algorithm is built for the swarm task. In Section 5, the problem model is solved and verified by experimental design, and the results of adaptive scheduling are given by comparing with the original scheduling algorithm. Finally, some suggestions for further research are discussed based on the current work Figure 2.

## 2. The Fourth-Directed Motif

A motif is a subnetwork structure that appears repeatedly in a particular network or in different networks and can reflect the functions that can be effectively implemented within a framework. In the early research, it is considered that the operational measurement of combat effectiveness is a very challenging task due to the large amount of complexity presented by the rich combat environment. In the literature [18,19,20], a new motif based combat effectiveness measurement method is proposed. According to the characteristics of combat mission and network motif, two kinds of motifs are abstracted from the combat network: an independent attack motif and joint attack motif. Then, Liu Jiajie from the National University of Defense Technology proposed the construction of a UAV swarm motif [21] from the perspective of function, and established six types of motifs: reconnaissance motif, attack motif, observation and attack motif, investigation and control and attack motif, and investigation-control and attack evaluation motif. Based on the idea of motif, this paper introduces motif into the role assignment mechanism of swarm organization reconstruction.

Firstly, according to the frequency of the motif, the basic motif is determined from the role assignment of swarm organization reconstruction, and it is regarded as the basic swarm coordination structure unit. Secondly, swarm organization reconstruction is a large-scale group cooperation. Due to the scale-free nature of the motif, small-scale swarm network collocation can be designed for large-scale swarm role assignment. Finally, the motif has a simple foundation, the internal structure of the motif and the interaction structure between the motif is simple [22,23,24], and the number of connections is small. For the communication connection between individuals restricted by geographical environment, weather and climate, the motif is suitable for the design of swarm level operations. On the other hand, the swarm access control mechanism directly allocates and authorizes roles from resources to roles, as shown in Figure 2, which does not effectively explain how swarm roles are dynamically allocated.

In this paper, four basic motifs with a simple foundation, a frequent occurrence, and practical operational significance are abstracted from the swarm cooperative operations network, and the motif model of swarm organization reconstruction is established. The four motifs-based coordination units are as follows:

Character update motif: The swarm leader platform updates the target list of new configurations based on the switching rule. Based on its functionality, this motif is named the update motif. At the same time, for convenience, the updated motif is denoted as *a*.

Role assignment motif: The swarm leader platform is assigned according to the alternative mode of switching rules. Based on its functionality, this motif is named the allocation motif. At the same time, for convenience, the assigned motif is denoted as *b*.

Role evaluation motif: The swarm leader platform evaluates the alternatives. Based on its function, this motif is named the evaluation motif. At the same time, for convenience, the evaluation motif is denoted as *c*.

Character generation motif: The swarm leader selects a new role assignment scheme and sends it to the subordinate platforms. Based on its function, this motif is named the generation motif. At the same time, for convenience, the character generation motif is denoted as *d*.

The definitions of closed and non-closed fourth order motif subgraphs in fourth order motif subgraphs are established. The flexible mechanism of swarm organization reconstruction is generated from the scheme of a swarm leader platform, denoted as directed network G=<V,E>, where *V* is the set of nodes, and *E* represents the set of edges. $E_{x}y$ represents the directed edge from the node *x* to *y*.

**Theorem** **1.**
*Closed fourth-order die body subgraph for a given fourth-order die body subgraph $G_{i}$, if there are edge sets $e_{ab},e_{bc},e_{cd},e_{da}\in E$ and $G_{i}$ for closed fourth-order die body figure.*


**Theorem** **2.**
*Nonclosed fourth-order motif subgraph for a given fourth-order motif subgraph $G_{j}$; we call $G_{j}$ a nonclosed fourth-order motif subgraph if there exists a set of edges $\exists e_{da}\notin E$.*


The basic structure of the flexible mechanism of swarm tissue reconstruction is set as Figure 3, Figure 4 and Figure 5. Non-closed fourth-order motif subgraph. For a given non-closed fourth-order motif subgraph, the motif a,b,c,d∈N, the connected edges of a single motif only contain four possibilities: forward, reverse, bidirectional and non-connected. As shown in Figure 4, $\exists e_{ac},e_{bd},e_{da}\in E$, $Num\left(e_{ac}\right)=Num\left(e_{bd}\right)=Num\left(e_{da}\right)=4$. It can be concluded that there are $C_{4}^{1}C_{4}^{1}C_{4}^{1}=64$ kinds of connection modes of closed fourth-order motif subgraphs, which means that there are 64 kinds of scheme outputs. Similarly, $e_{da}=null$, which is a nonclosed fourth-order modular subgraph, has $C_{4}^{1}C_{4}^{1}C_{4}^{1}=64$ types of connections, so all of the evolution models have nine types of evolution, and there are a total of $2^{9}$ types of connections. The adjacency matrix of the network is denoted as $C={\left[C_{ij}\right]}_{n\times n}$, which can be expressed as: (1)$$c_{ij}=\left\{\begin{array}{cc} 0; & i=j\\ 1; & i\neq j \end{array}\right.$$

If nodes are connected, the value is 1; otherwise, the value is 0.

## 3. The Index of M-SAR2

Because directionality determines the complexity of edges and nodes, the variants of the fourth-order directed closed motif and the fourth-order directed non-closed motif are different [25], which need to be considered in Figure 4, respectively. In order to measure the effect of model body mass structure change [26], three indicators should be established for analysis, and dynamic role rematching mainly depends on three indicators:Analyzes the structural similarity from the importance of the motif (ASS);Quantifies the importance of the motif from a statistical point of view (QIBF);Assigns a node to the earliest processing machine that causes it to start (ANMMT).

The task completion time ($T_{m}ct$) is set to: (2)$$T_{m}ct=maxFin_{T}.$$

$Fin_{T}$ is the set of times when all task changes are completed. in the fourth-order directed motif; let $a_{l}^{i}$ be the adjacency matrix of *i*. In order to distinguish directionality, in-degree and out-degree are introduced. Thus, for *i*, node degree is divided into the total degree of $k_{total}\left(i\right)$ and obtains $k_{in}\left(i\right)$, leaving $k_{out}\left(i\right)$, including: (3)$$k_{total}\left(i\right)=k_{in}\left(i\right)+k_{out}\left(i\right).$$
(4)$$k_{in}\left(i\right)=\sum_{j\in V(i)}a_{ji}.$$
(5)$$k_{out}\left(i\right)=\sum_{j\in V(i)}a_{ij}.$$*j* denotes the adjacency of *i*, and V(i) denotes the set of *j*. When k⩽10 is used for the directed weighted network model established in the experiment, the distribution of network input-in degree is a gentle process. For a network whose degree distribution follows Poisson distribution, the degree distribution function is set as: (6)$$p\left(k\right)=\lambda^{k}e^{-k}/k!.$$

To calculate the association degree of nodes, the following parameters are defined: $T_{ij}^{in}$: Node *i* and node *j* have similar in-degree trust; $T_{ij}^{out}$: Node *i* and node *j* have similar outdegree trust; $T_{ij}^{z_{in}}$: Denoted as the normalized $T_{ij}^{in}$; $T_{ij}^{z_{out}}$: Denoted as the normalized $T_{ij}^{out}$; $T_{ij}$: Denoted as the weighted trust degree of node *i* and node *j*. Jaccard similarity coefficient is used to define $T_{ij}^{in}$ and $T_{ij}^{out}$,
(7)$$T_{ij}^{in}=\frac{|{e_{ij}\left(i\right)}_{in}\cap {e_{ij}\left(j\right)}_{in}|}{|{e_{ij}\left(i\right)}_{in}\cup {e_{ij}\left(j\right)}_{in}|+1}$$
(8)$$T_{ij}^{out}=\frac{|{e_{ij}\left(i\right)}_{out}\cap {e_{ij}\left(j\right)}_{out}|}{|{e_{ij}\left(i\right)}_{out}\cup {e_{ij}\left(j\right)}_{out}|+1}$$
where ‘∣’ represents the number of elements in the set within the symbol. To avoid the situation in which the denominator is zero after the subtraction of the out degree and in degree of the motif, add 1 to the denominator and then standardize it: (9)$$T_{ij}^{z_{in}}=f\left(\left(T_{ij}^{in}-\mu_{i}^{in}\right)/\sigma_{i}^{in}\right)$$
(10)$$T_{ij}^{z_{out}}=f\left(\left(T_{ij}^{out}-\mu_{i}^{out}\right)/\sigma_{i}^{out}\right)$$

Among them, the $f\left(x\right)=\frac{e^{x}}{1+e^{x}},\mu_{i}^{in},\mu_{i}^{out}$ is all die bodies, respectively, $T_{ij}^{in},T_{ij}^{out}$, $\sigma_{i}^{in},\sigma_{i}^{out}$ is the standard deviation of all motif, respectively. Finally, the trust degree between node *i* and node *j* is obtained by a weighted average of the two similarity trust degrees: (11)$$T_{ij}=\alpha T_{ij}^{z_{in}}+\left(1-\alpha \right)T_{ij}^{z_{out}}$$

The 0<α<1 is the weighting coefficient.

## 4. The SORAM Assignment Algorithm

### 4.1. Swarm Role Rematching Mechanism

The swarm role reassignment mechanism is implemented by the swarm leader platform at the swarm layer. It is a task adaptation mechanism in the task domain. Its main goal is to carry out online adaptive planning and flexibly reassign swarm roles to support the swarm task adaptation according to the new internal state or new task requirements of the swarm. Departure conditions are the following two types of events: the superior issues a new task or the internal cooperation status is found to be faulty. The rule base includes task update rules, status detection rules, role assignment rules, role evaluation rules, and so on. The specific steps of the mechanism are as follows:The swarm leader platform produces and updates alternative new configuration modes according to the switching rules;The swarm leader platform is allocated according to alternative modes of switching rules;Evaluation of alternative modes of related platforms of swarm leaders;The leader platform of the swarm selects a new role allocation scheme and sends it to the subordinate platforms.

It belongs to a small number of centralized and distributed structures. The mechanism must meet the following preconditions: Online role authorization reassignment. The mechanism execution needs the support of the online swarm role assignment algorithm, which can automatically generate the role assignment scheme based on the updated task list or state events in a short period, and task or cooperation compatibility judgment in advance. As a part of command and control, this mechanism can be programmed into the function of swarm organization reconfiguration of hardware and software supporting swarm leader platform. The adjacency matrix considers the correlation degree of nodes and analyzes the structure of a directed graph from a quantitative perspective. Therefore, the fourth-order motif can be mapped into the adjacency matrix and be solved by the algorithm:(12)$$A.arcs\left[i\right]\left[j\right]=\left\{\begin{array}{cc} W_{ij}; & <vi,vj>or(vi,vj)\in VR\\ +\infty ; & else \end{array}\right.$$

In the adjacency matrix of a digraph:

The meaning of the *i* line: the arc ending with the node vi (i.e., the exit edge).

The *i* column meaning: node vi as the beginning of the arc (that is, in degree edge).

Analysis 1: The adjacency matrix of a digraph may be asymmetric.

Analysis 2: The outdegree $k_{out}\left(i\right)$ of a vertex is equal to the sum of the elements of the *i* row.

The entry degree of a vertex $k_{in}\left(i\right)$ is equal to the sum of the elements in the *i* column Vertex degree $k_{total}\left(i\right)$ is equal to the first *i* line elements plus the sum of the sum of the first *i* column elements. The adjacency matrix of the weight graph is expressed as Equation (12).

### 4.2. Swarm Reconstruction Scheduling Allocation Algorithm

Scheduling problem in swarm is a core problem of role assignment in swarm organization reconstruction. The multi-dimensional dynamic scheduling method is often used for equipment scheduling in war and has achieved good results [27,28,29,30,31]. By improving the multi-dimensional dynamic scheduling method, the role assignment algorithm for swarm reconstruction is designed. The logical structure is shown in Figure 6 below:

Therefore, in this study, the swarm reconstruction role assignment Algorithm 1 is used to determine whether there is a new task requirement and update the task motif given a basic organizational structure of the fourth-order non-closed-loop motif. The new mandate considers two scenarios:

1. The structure of the original fourth-order non-closed motif does not meet the requirements of the swarm due to uncertain factors.

2. The basic fourth-order motif structure is disturbed by phenomena such as node interruption.

Among them, this paper believes that node interruptions such as uncertain factors and disturbances of swarm task links can be handled by flexible variation (as shown in Figure 7), and the nonlinear mixed Boolean programming method is used to establish mathematical models to optimize the risks.

A reconfigurable flexible network (Reconfigurable network) is a kind of variable network system that can meet the changing task requirements by adjusting the function, performance, or corresponding relationship of network nodes through reconfigurable technology. To meet the requirements of battlefield adaptability, the network organization structure must be variable.
**Algorithm 1** Swarm reconstruction scheduling allocation.initialize: $T_{F},T_{S}=null,Fin_{T}=null$; Task case**1**: $f=min_{ft\in Fin_{T}}\left(ft\right)$ ( $Fin_{T}$ )**2**: $Fin_{T}\leftarrow Fin_{T}\;f$**3**: $t_{t}otal\leftarrow t_{t}otal\cup f$ ($t_{t}otal$ )**4**: $T_{f}\leftarrow T_{f}\cup F_{G}$**5**: $Spare\leftarrow Spare+m(F_{G})$**6**: READY Update**7**: IF: $\forall k\in READYcannotsatisfyD\cdot m{\left(T_{i}\right)}^{\prime }\leq Spare$$T_{i}$**8**: update p(k)**9**: Go to Line 1.**10**: ELSE: GotoLine12.**11**: ENDIF**12**: IF: $\forall i\in READYcannotsatisfyT_{ij}∼T_{ij}^{in},T_{ij}∼T_{ij}^{out}$,**13**: $T_{S}\leftarrow T_{ij}\cup i$.**14**: $Fin_{T}\leftarrow Fin_{T}\cup \left(f+t_{i}\right)$**15**: Select $max\left(T_{ij}\right)fromT_{S}$**16**: spare $\leftarrow spare-m(F_{G})$, ∀i∈READY**17**: DOUNTILL: $D\cdot m{\left(T_{i}\right)}^{\prime }\geq Spare$**18**: Go to Line 12.**19**: ENDDO**20**: IF: $M\cdot m{\left(T_{i}\right)}^{\prime }\geq Spare,\forall i\in READY$**21**: Go to Line 1.

Line 1 is a collection of completion times of ongoing tasks; Line 3 is the set of completion times of all tasks, and the Line 6 is a collection of tasks that have been prepared.

There are many clustering intelligence algorithms, such as the Artificial Bee Colony (ABC). Its main purpose is to realize the sharing and communication of bee colony information, in order to find the optimal solution to a problem. However, our fourth order die body to reconstruct the role of each user does not need to share information, has dynamic adjustment in the cluster motif, information needs storage space, and the loading plan also requires a short amount of time. Thus, it is a multi-objective optimization problem. In this context, the NSGA-2 used the elite search strategy, rapid non dominated sorting, and has more range of motion.

### 4.3. Improved NSGA2 Based on Swarm Reconfiguration (SR-NSGA-2)

The cluster organization reconfiguration mechanism design problem is a multi-objective optimization problem. To solve the task planning optimization problem that meets the three objectives, multi-objective evolutionary optimization (MOEA), called NSGA-2, has better performance in the same algorithm [32,33]. It has the better performance with fewer iterations.

For a multi-objective optimization problem, the optimal solution of the problem may be more than one but should be a group, and we usually call this group of optimal solutions a non-dominated solution set of the corresponding multi-objective optimization problem, or a Pareto solution set, where each solution is called a Pareto solution. There are many methods to solve multi-objective optimization problems, such as the common goal programming method, goal decomposition method, the goal more to less method (expressing multiple goals as one), etc.

The SORAM is a multi-objective optimization problem, including the scheduling time of the motif, and the stability and similarity of the structure of the motif, which can be quantified as the capability index in the case. This kind of cluster task planning problem is a small number of centralization combined with a discretization problem, that is, a small number of nodes are control platforms, and other platforms are discrete and mobile. The distributed problem means that the unit can control its actions and make its own decisions, each of which is optimal to maximize the overall benefit. This is a very difficult global optimization problem. In the ARD problem, the commander needs to assign tasks to each UAV, and then discrete UAVs make their own decisions to obtain a better action plan. Finally, the overall benefit is greater than the sum of the parts.

Task planning is a multi-objective problem, which optimizes two objectives at first. Slowly, the number of targets increases, a non-dominated ranking emerges, and slowly the set of preference decision-makers emerges in the search process and then coevolves. Moreover, it has been proved that the clock counting algorithm is better, such as NSGA-2, HYPE, MOEA/D, EMOEA, etc. [34,35,36]. Evolutionary algorithms have natural advantages in solving multi-objective problems. An evolutionary multi-objective optimization algorithm can optimize multiple objective functions at the same time, and output a set of non-dominated Pareto solution sets so that it can effectively solve multi-objective problems.

#### 4.3.1. SR-NSGA-2 Algorithm Architecture

In this study, we found that the distribution of individual neighbors directly determines the density of the surrounding space. However, most papers rarely consider the influence of individual neighbors on population diversity in the process of evolution. Thus, we use the crowding distance metric. In most other evolutionary algorithms, individuals in the middle group are ranked as non-dominated with the next-generation population, while SR-NSGA-2 is based on the NSGA-2 algorithm. After the first non-dominated ranking, a new crowding degree calculation is introduced to update the neighborhood, as shown in Figure 8, and then the population iteration is performed. Compared with the original NSGA-2, SR-NSGA-2 has the advantage of a faster solution, which means that it can find a better Pareto solution with fewer iterations: (13)$$Crowd_{i}=\sum_{i=1}^{M}\left(\right|g_{k}\left(x_{i-1}-g_{k}\left(x_{i+1}\right)\right)\left|\right)$$
(14)$$LON\left[Z_{i}\right]=\sum_{i=1}^{M}\left(|g{\left(Z_{i+1}\right)}_{M}-g{\left(Z_{i-1}\right)}_{M}\right)|/\left(g_{m}^{max}-g_{m}^{min}\right)$$

In order to ignore the difference between the range of different objective functions, it is also necessary to divide by the endpoint difference in the hierarchy. It can be concluded that, when there are K objective functions, the quarterly distance calculation formula of *Z* between individuals can be expressed as Equation (Equation 14) [37], where $g_{m}^{max}$ is the maximum value between populations in m objectives, and vice versa.

The SR-NSGA-2 adopts the research framework of NSGA-2 and consists of population initialization, population evolution, genetic iteration, and selection. In particular, SR-NSGA-2 introduces a new crowding distance calculation method, which considers the influence of neighborhood on subsets (Figure 8 and Figure 9).

We added Equation (Equation 14) to add the crowding calculation after the fast non-dominated sorting, as shown in Figure 9; comparing NSGA-2 and SR-NSGA-2, it has a faster convergence rate, and it changed the order of cross-mutation. Next, we use the SR-NSGA-2 algorithm to solve the role reconstruction problem.

#### 4.3.2. Initial Population Generator

The optimization process of SR-NSGA-2 is shown in Algorithm 2. We need to ensure that the number of input population follows the priority of the initial and final nodes of the task, and generates a random number of task sequences. Firstly, the two populations are divided into N/10 equal parts, and the population functions are initialized as pt0 and fpt0 Figure 10.

Select the tasks that are ready shown as Algorithm 2. All reconstructed roles are simply arranged using random numbers. First, a ready role is selected, and this role is selected first. Then, the selected role is removed to update the motif task link by Equations (2) and (11). Then, a ready task is selected, and the priority task link is updated to obtain a solution after all the tasks are selected. In this case, the mandatory relation/priority it satisfies is the order of selecting the motif with higher trust.
**Algorithm 2** SR-NSGA-2 generation.**1**: Define the scale of population and $iter_{max}$**2**: The interval [a, b] is divided into N/10 equal parts,**3**: Pt0=unifrnd(minx,maxx,N,numvariate);**4**: fPt0=zeros(N,numfun+2)**5**: Pt0=cat(2,Pt0,fPt0)**6**: END

#### 4.3.3. Crossover and Mutation Operator

We use crossover fragments to cross over the parent population, and the subpopulation iterates through the genes after crossover. The operator of crossover generation meets the requirements because the parent chromosome conforms to all the task chains. Firstly, a pair of chromosomes pr0 and fpr0 were selected from the population, and the genes were divided into two partial acids at the emphasis of ui. Gene segment 1 was from 1 to [ui/2]−1, and gene 2 was from [ui/2] to ui. The new chromosome structure was as follows. The crossover operator of the fragment still keeps the constraint of the parent operator (Algorithm 3). According to the constraints, the crossed chromosome segments will be rearranged, and the preferential relationship will be strictly executed in the offspring.

Do not cross the mutation, to avoid the population of the same number of individuals, so as to maintain the diversity of the population. The order of crossover followed by mutation was used, and pMutation=0.05 was set. The optimization process of crowding distance assignment is shown in Algorithm 4. The mutation operator uses the two-point exchange method to randomly select a point from the chromosome to see whether it meets the exchange. If the position of the exchange point is in the upper and lower order task with a forced relationship, no mutation operation will be carried out. Then, a random point will be opened to see whether it meets the requirements. For example, the task set with the forced relation has T4−T1−T2. If the ‘4’ is randomly selected and the set is found, the above task set will not mutate, and then a mutation factor is obtained at random.
**Algorithm 3** Crossover and mutation operator.**1**: Define the corssPt and Pmu**2**: for j=1:numvariate**3**: crossPt(:,i)=randperm(N)**4**: for i=1:ceil(N/2)**5**: if (pccross < pc)**6**: update Qt1 and Qt2**7**: elseif**8**: $deta1\leftarrow {\left(2*pmu1\right)}^{(}1/\left(cetam+1\right))-1;$**9**: $deta2\leftarrow {\left(2*pmu2\right)}^{(}1/\left(cetam+1\right))-1;$**10**: end**11**: end

**Algorithm 4** Crowding distance assignment.
**1**:Input: Fi of size 1∗(2∗N+1)
**2**: Define NFi=Fi(1,1); Idistance=zeros(1,NFi)
**3**: if NFi<=2
**4**: Idistance=Inf∗ones(1,NFi);
**5**: else b=Fi(2:(Fi(1,1)+1));
**6**: fprintf(’→ Idistancel= **7**: for i=1:numfun
**8**: sort(funvalueI1(:,i))←[valueI1,rankI1]
**9**: for j=2:(NFi−1)
**10**: Idistance(rankI1(j)) ← Crowded distance indicator
**11**: end


## 5. Case Study

### 5.1. The Experiment Design

Taking the fire strike system composed of a single unmanned swarm as a case, a task planning scheme is designed to automatically generate the role allocation algorithm for cluster reconstruction to verify the effectiveness of the SORAM method. Design case parameters: the red side is mainly composed of unmanned vehicles, UAVs, and patrol missiles to strike a mobile variable target, and the use case has 15 combat missions T1,T2,.... The available unmanned combat units are shown in Table 1. Capabilities in 5, 4 systems, and 5 combat units are selected to perform ARD missions. Set parameters is based on Algorithm 1 considering time and cost. The calculation of the trust degree can be obtained according to Formula (11) in the index design. See Algorithm 1 for the specific pseudocode of the algorithm.

### 5.2. Solution of a Motif Fitting Scheme

It is known that the current ARD execution cost is shown in Table 1. War experts estimate that the time to complete each task according to the difficulty of the task, the types, and quantities of the molds required for the execution of the task, and combined with battlefield conditions, as shown in Table 2. The scheme of the swarm role reconstruction mechanism is shown in Figure 11.

The available UAV resources are 3 relay UAVs, 20 decision UAVs, 15 reconnaissance UAVs, 18 attack UAVs, and 9 surveillance and combat UAVs. After analyzing task elements such as task difficulty and the number of enemy targets, the capability requirements and communication requirements of the mission are quantified [13,38], the demand vector of the mission’s motif is calculated, and their completion time is estimated, as shown in Table 2. Select a feasible task priority execution order, based on a MATLAB dynamic list scheduling method, and output the basic task start time and end time.

#### Results Analysis

Taking this order as the input, the main purpose is to design the disturbance point to observe whether it is quickly adaptable. At this time, SORAM is used to calculate the set Ts and total in the case of unexpected changes, and the Gantt graph is used to represent the scheduling difference Figure 12.

As shown in Figure 13, given the disturbance point U1 and the appearance of the new strike task, U3 changed rapidly and cooperated with U4 to complete the task scheduling. According to the above experimental results, SORAM is helpful for the rapid adjustment of the mission link of the UAV swarm to complete the mission when the UAV swarm performs the mission in the face of the unexpected situation of organization mobilization or combat mission. In this simulation experiment, the method of experimental design is adopted to verify the scheme of the swarm role redistribution mechanism, and the unexpected node is designed. When the unexpected node occurs, U3 and U4 quickly adjust the task in a short time, and the effectiveness of the scheme of the redistribution mechanism is preliminarily obtained.

All UAVs will disconnect the motif link after the mission, and the conditional link motif will be triggered at the beginning of each mission at an endpoint. Being free to choose a role assignment scheme, according to the roles and tasks, gives a platform to new roles, through the code to perform the task, a task to encode in ARD, application of SR-NSGA-2 methods to choose clustering task planning for an unmanned aerial vehicle, using a SORAM mechanism, and automatic generation of different mission planning strategy. The improved NSGA-2 algorithm of crowding degree ranking was applied for rapid iteration and convergence, and some non-subjective task planning schemes were obtained. After 50 iterations, two groups of comparative experiments were conducted, and the results showed that SR-NSGA-2 only took 4.574847 s, and the second group of SR-NSGA-2 only took 4.574847 s. This is 0.035494 s faster than NSGA-2’s 4.610341 s. Compared with the two non-dominated sorting algorithms, SR-NSGA-2 has more Pareto front than NSGA-2 for this SORAM problem.

After preliminary comparison with 50 selected samples, it is found that the operator distribution of the improved SR-NSGA-2 is better than that of NSGA-2. Later, with the increase in the number of iterations, the operator distribution has a big difference Figure 14, Figure 15, Figure 16 and Figure 17. Figure 14 shows the 3D comparison of three indexes of the two algorithms. (a–d) are the results of 50, 100, 500, and 1000 iterations, respectively. It can be seen that SR-NASG-2 gradually forms a scale and is closer to the center point set. In the ASS and QIBF sections Figure 15, we found that SR-NSGA-2 can maintain a relatively good distribution even when the completion time is short. In ANMMT and ASS sections Figure 16, SR-NSGA-2 can maintain a relatively stable use time while maintaining good structural similarity. Under the section of QIBF and ANMMT Figure 17, the distribution of SR-NSGA-2 does not tend to a certain side, while NSGA-2 is more inclined to the index of structural similarity.

We set different criteria and calculate the number of task planning schemes by using two algorithms to meet these criteria. In most of the criteria, SR-NSGA-2 is better, and more task planning schemes following the dynamic role allocation mechanism scheme can be obtained.

## 6. Conclusions

This paper designs and implements the implementation method of link reconstruction of a UAV swarm mission in a dynamic battlefield environment. It goes through three stages: fourth-order module design and modeling, index design, and mechanism algorithm design. According to the new status or new task requirements in the swarm, online adaptive planning is carried out, and the roles of the swarm are redistributed to support the task adaptation and achieve the combat objective. In the motif design stage, using the idea of dynamic adjustment of edges and nodes of a complex network, the fourth-order motif evolution is designed to lay the foundation for link reconstruction. In the index design stage, the importance of the motif is quantified from the perspective of statistics, and the trust degree of the structure is defined by the Jaccard coefficient. In the design stage of the algorithm, the adjacency matrix is used to abstract the motif, and the multi-dimensional scheduling algorithm is improved to solve the task planning scheme. To quickly obtain the scheduling scheme, the traditional NSGA-2 method is improved for the SORAM problem, and the SR-NSGA-2 algorithm is established. Using the same example conditions, it is found that SR-NSGA-2 is significantly better than NSGA, and it has faster-solving speed and can quickly obtain the task adaptive planning scheme. The experimental results show that the method can effectively face unexpected situations in the process of performing the task and has strong adaptability. This method can not only be applied to the adaptive design of UAV swarm mission planning but also has important significance in the adaptive design of swarm system mission planning.

The rapid cluster reconfiguration method presented in this paper can be applied to the problem of prior planning in the task allocation process of swarm operations. In addition, it can also be applied to other typical scenarios, such as fire rescue, forest rescue, and other scenarios, and the task allocation problem of fact UAV swarm detection. The future scope of the method can be extended as follows:

1. Strengthen the scheduling algorithm and analyze the scheduling process more carefully.

2. Propose the combination of a scheduling algorithm and genetic algorithm, hoping to find a more appropriate solution method and optimize the solution quality in the next step.

3. Design a more suitable simulation scenario, covering the problem of real-time state change.

In further work, combined with the simulation scenario, a better flexible mechanism is designed to solve the real-time scheduling problem in a large-scale dynamic environment and improve the adaptability and robustness of the cluster. 

## Figures and Tables

**Figure 1 sensors-22-08799-f001:**
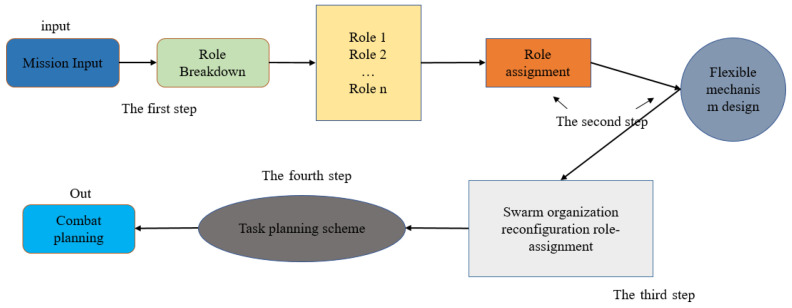
The swarm organization reconfiguration role-assignment planning.

**Figure 2 sensors-22-08799-f002:**
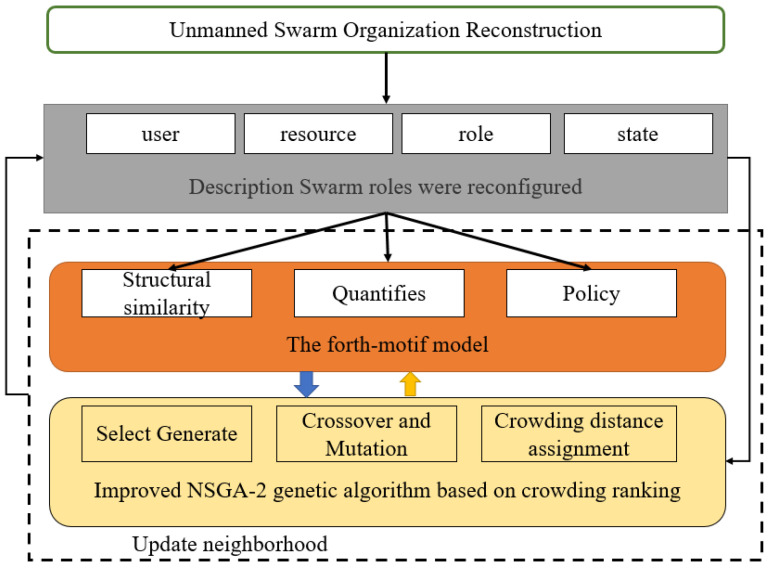
Dynamic role distribution positioning.

**Figure 3 sensors-22-08799-f003:**
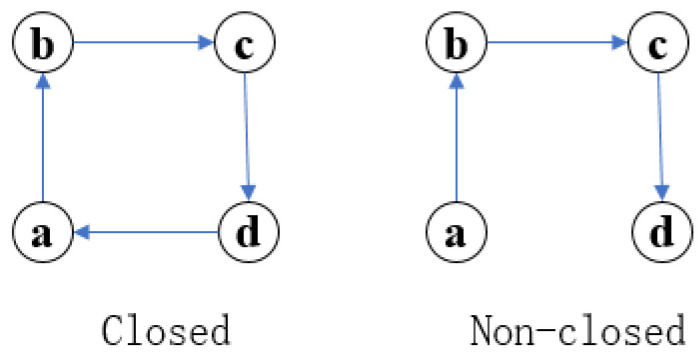
Closed fourth-order motif subgraphs and non-closed fourth-order motif subgraphs.

**Figure 4 sensors-22-08799-f004:**
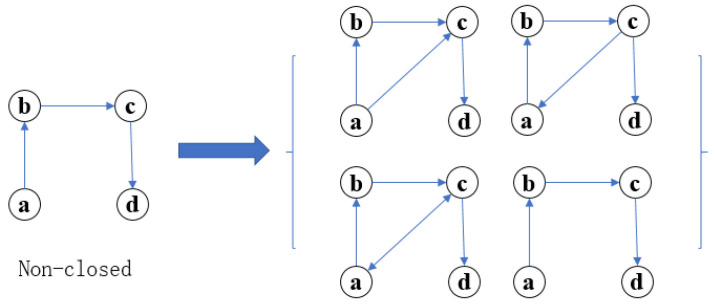
Diagram of motif evolution.

**Figure 5 sensors-22-08799-f005:**
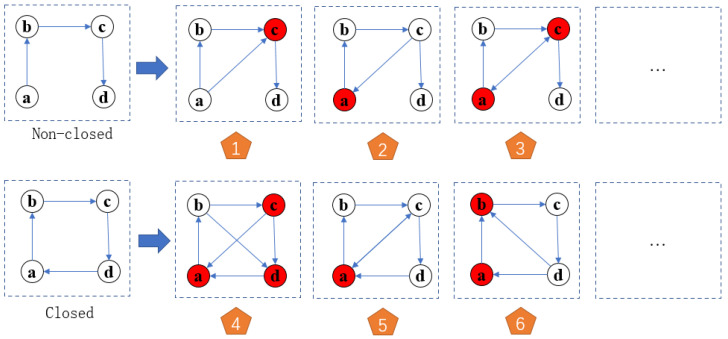
The motif evolution reconstruction.

**Figure 6 sensors-22-08799-f006:**
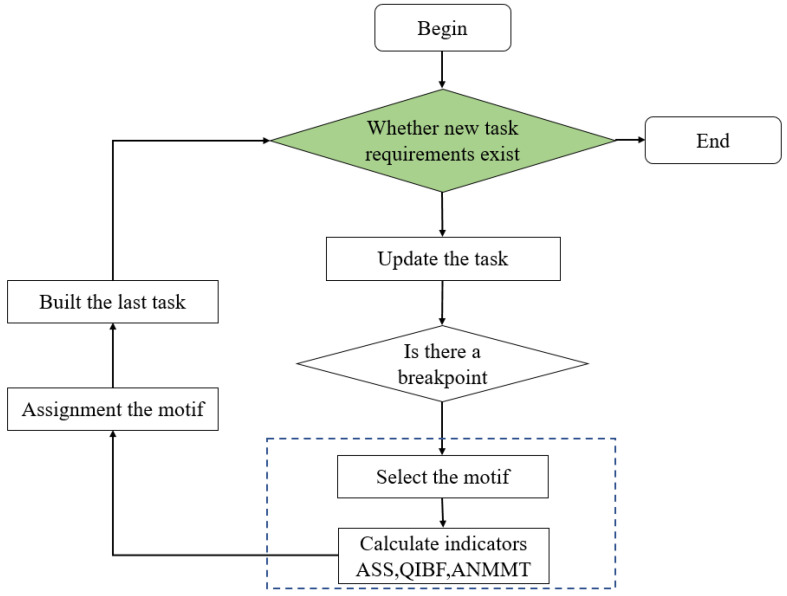
The swarm reconstruction scheduling allocation algorithm.

**Figure 7 sensors-22-08799-f007:**
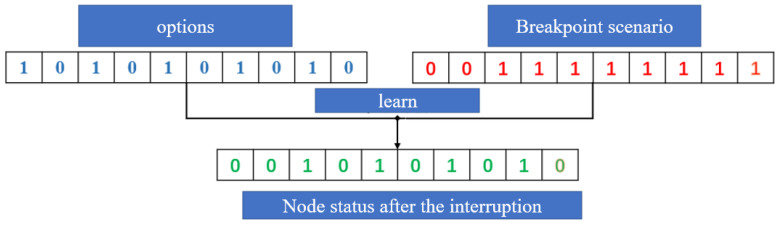
Interrupt point interference.

**Figure 8 sensors-22-08799-f008:**
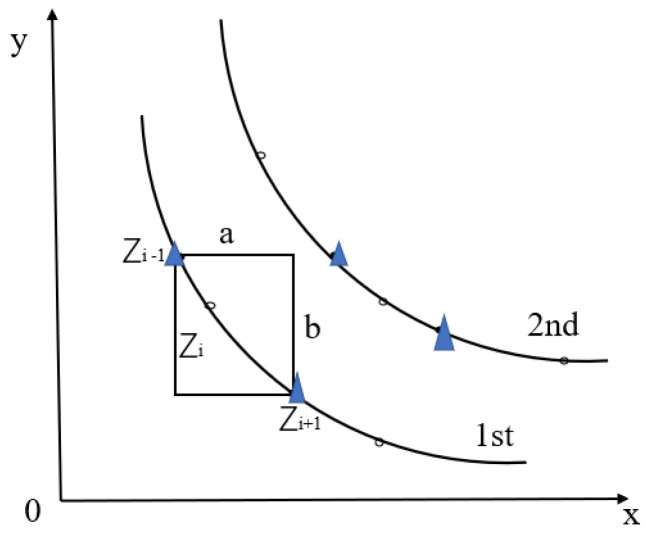
Individual crowding calculation.

**Figure 9 sensors-22-08799-f009:**
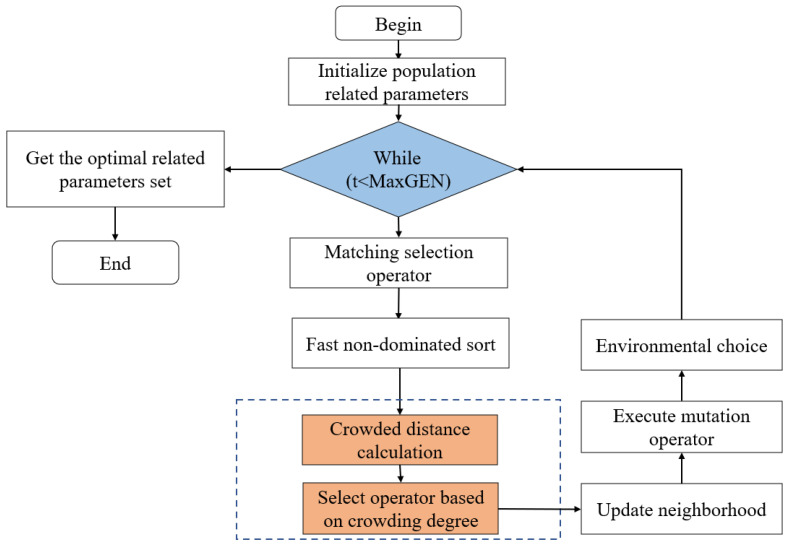
The framework of an SR-NSGA-2.

**Figure 10 sensors-22-08799-f010:**
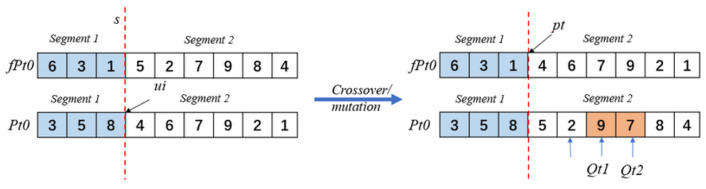
Population selection fragment.

**Figure 11 sensors-22-08799-f011:**
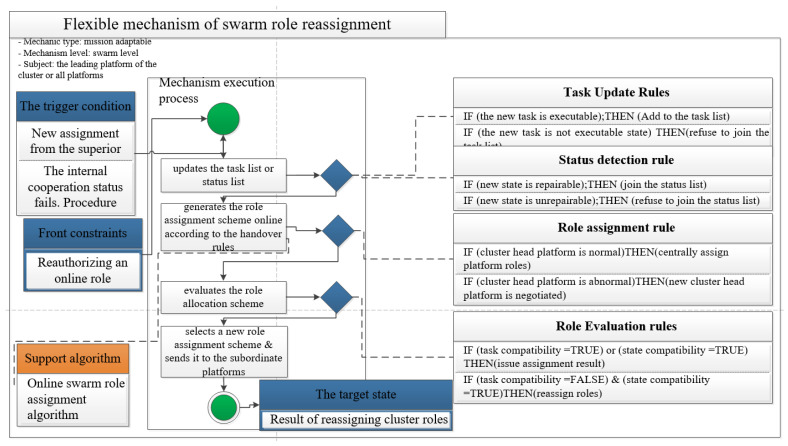
Swarm role reconfiguration mechanism scheme model.

**Figure 12 sensors-22-08799-f012:**
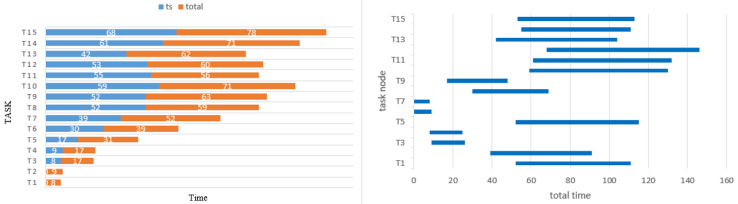
The Gantt of task planning.

**Figure 13 sensors-22-08799-f013:**
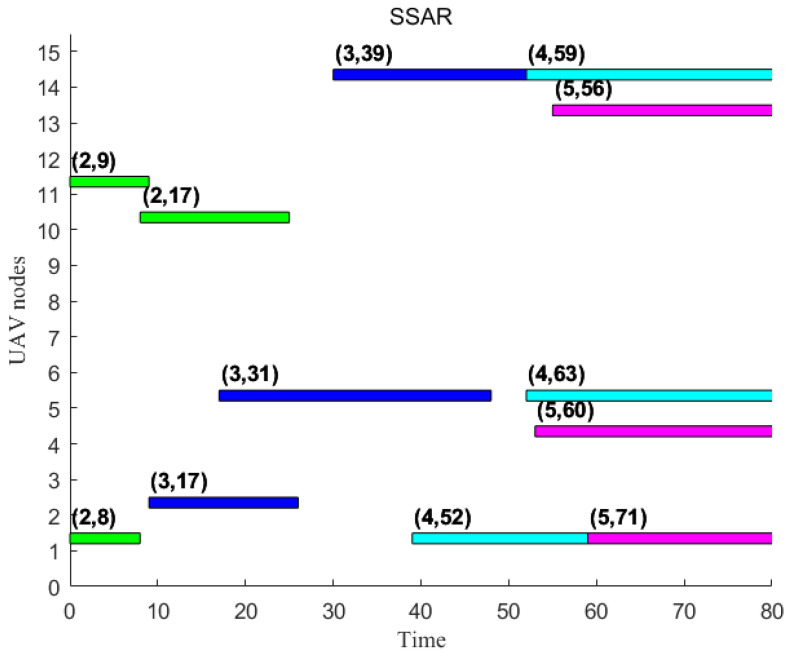
The Gantt comparison of task planning.

**Figure 14 sensors-22-08799-f014:**
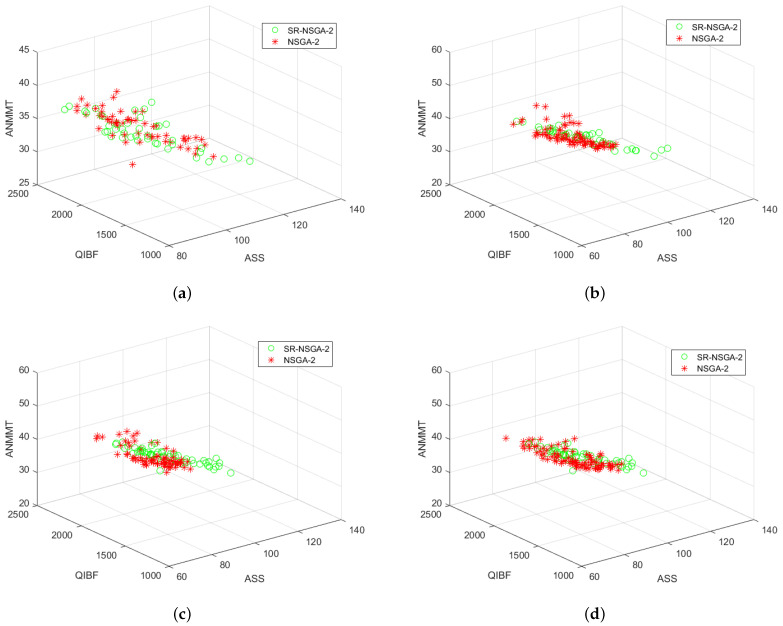
Operator iteration of NSGA-2 and SR-NSGA-2 for the SORA problem.

**Figure 15 sensors-22-08799-f015:**
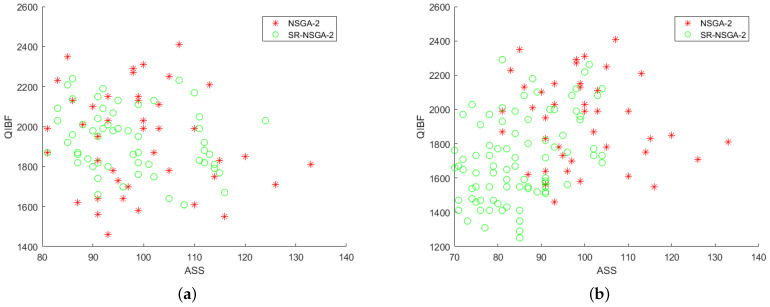
Index analysis chart of ASS and QIBF.

**Figure 16 sensors-22-08799-f016:**
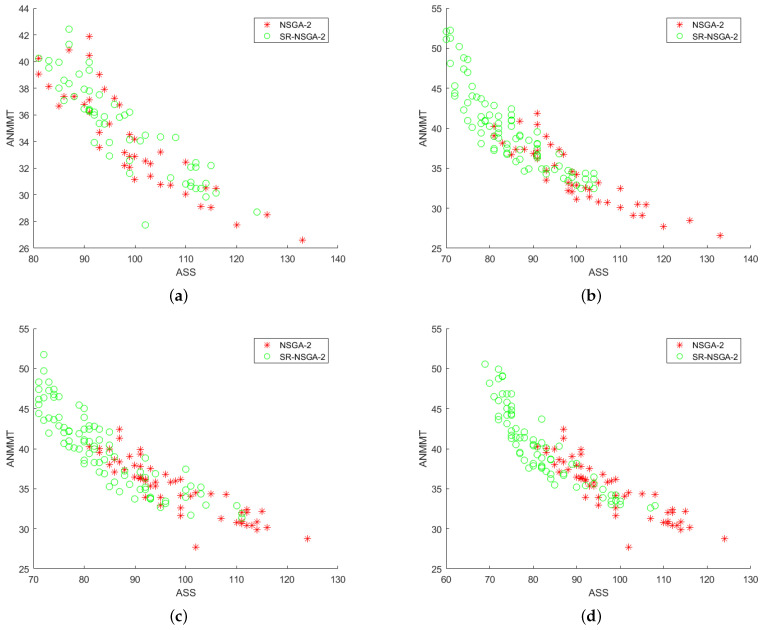
Index analysis chart of ASS and QIBF.

**Figure 17 sensors-22-08799-f017:**
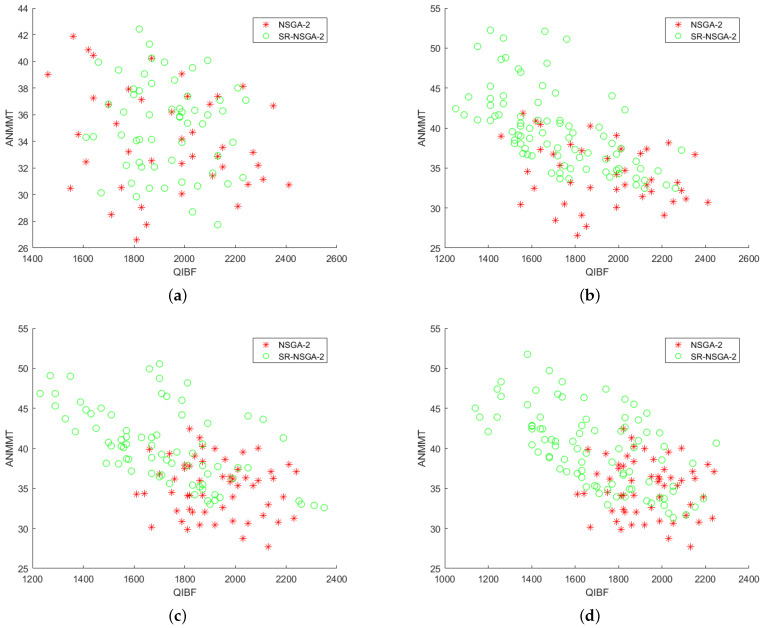
Index analysis chart of ASS and QIBF.

**Table 1 sensors-22-08799-t001:** Swarm performance unit.

Activity	Num	Ability	N	Demand	System	Cost
position	1	precise positioning	1	20	unmanned combat system	30K
early warning	2	air patrol	2	20	cruise missiles	120K
		fast search	3		armored vehicles	60K
		tracking	3		unmanned ground vehicle	40K
confirmation phase	3	high speed search	5	15	UAV	45K
		Fire monitoring	5		airship	45K
		Fire suppression	5		satellite	28K

**Table 2 sensors-22-08799-t002:** Task information sheet.

Motif	T1	T2	T3	T4	T5	T6	T7	T8	T9	T10	T11	T12	T13	T14	T15
a	3	0	2	1	3	2	2	2	1	3	2	0	1	0	1
b	3	1	1	2	1	0	3	1	3	1	2	0	0	1	3
c	3	1	0	1	0	1	2	1	2	2	3	2	1	0	2
d	2	0	1	0	1	2	1	1	1	2	0	2	1	2	0
Time(Min)	15	8	7	12	6	11	10	8	18	7	9	7	10	11	14

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
