# Peer review of "Role Assignment Mechanism of Unmanned Swarm Organization Reconstruction Based on the Fourth Directed Motif"

_sensors, 2022, doi:10.3390/s22228799_

Round 1

Reviewer 1 Report

This article is about Role Assignment Mechanism of Unmanned Swarm Organization Reconstruction Based on The Fourth Directed Motif. The overall design of the article is very good and the details are well explained. However, there are details in this article that, if corrected, can increase the quality of the article:

1- In the summary part of the article, the methododolgy is explained extensively , but no explanation is given about the results obtained.

2- Figure 8 needs correction 

Reviewer 2 Report

This paper proposes a role assignment model for organizational reconfiguration at the swarm layer.

The manuscript is well structured, but some important points need to be revised:

1)    The literature review should be improved considering the state-of-the-art methods. Without improved comparison, it is difficult to evaluate contributions;

2)    Throughout the introduction authors need to highlight their point-by-point contributions. Such information needs to be very clear to the reader;

3)    Some abbreviations and acronyms are not defined;

4)    Typos and grammatical errors could be noted;

5) The font size in Figure 11 is too small, and the picture clarity is insufficient;

6)    Figure 2 is not contextualized in the text. It's loose...;

7)    Improve the analysis of the results presented in Figures 14-17;

8)    Expand the conclusion to add feature works.

Reviewer 3 Report

The article is devoted to the actual interesting topic and will be interesting to the reader. The existing methods of solving the problem are well illuminated, adequate promising areas of development of this scientific and technical direction are shown. The material is presented clearly and understandable, illustrated well. The convincing practical implementation of the approaches considered is given. I believe that the article can be published without refinement.

Author Response

Thank you very much for your comments and recognition. Your recognition is very important to us, which makes us build confidence in the following large-scale swarm flexible reconstruction research in an uncertain dynamic environment.

Round 2

Reviewer 2 Report

The suggested revisions were implemented in the manuscript.